# Plasma Malondialdehyde and Risk of New-Onset Diabetes after Transplantation in Renal Transplant Recipients: A Prospective Cohort Study

**DOI:** 10.3390/jcm8040453

**Published:** 2019-04-04

**Authors:** Manuela Yepes-Calderón, Camilo G. Sotomayor, António W. Gomes-Neto, Rijk O.B. Gans, Stefan P. Berger, Gerald Rimbach, Tuba Esatbeyoglu, Ramón Rodrigo, Johanna M. Geleijnse, Gerjan J. Navis, Stephan J.L. Bakker

**Affiliations:** 1Division of Nephrology, Department of Internal Medicine, University Medical Center Groningen, University of Groningen, 9713 GZ Groningen, The Netherlands; manueyepes@gmail.com (M.Y.-C.); a.w.gomes.neto@umcg.nl (A.W.G.-N.); s.p.berger@umcg.nl (S.P.B.); g.j.navis@umcg.nl (G.J.N.); s.j.l.bakker@umcg.nl (S.J.L.B.); 2Department of Internal Medicine, University Medical Center Groningen, University of Groningen, 9713 GZ Groningen, The Netherlands; r.o.b.gans@umcg.nl; 3Institute of Human Nutrition and Food Science, Christian-Albrechts-University of Kiel, Herrmann Rodewaldstrasse 6, D-24118 Kiel, Germany; rimbach@foodsci.uni-kiel.de; 4Institute of Food Science and Human Nutrition, Department Food Development and Food Quality, Gottfried Wilhelm Leibniz University Hannover, Am Kleinen Felde 30, D-30167 Hannover, Germany; esatbeyoglu@lw.uni-hannover.de; 5Molecular and Clinical Pharmacology Program, Institute of Biomedical Sciences, Faculty of Medicine, University of Chile, Av. Independencia 1027, CP 8380453 Santiago, Chile; rrodrigo@med.uchile.cl; 6Division of Human Nutrition and Health, Wageningen University and Research, P.O. Box 47, 6700 AA Wageningen, The Netherlands; Marianne.geleijnse@wur.nl

**Keywords:** malondialdehyde, oxidative stress, new-onset diabetes, renal transplantation

## Abstract

New-onset diabetes after transplantation (NODAT) is a frequent complication in renal transplant recipients (RTR). Although oxidative stress has been associated with diabetes mellitus, data regarding NODAT are limited. We aimed to prospectively investigate the long-term association between the oxidative stress biomarker malondialdehyde (measured by high-performance liquid chromatography) and NODAT in an extensively phenotyped cohort of non-diabetic RTR with a functioning graft ≥1 year. We included 516 RTR (51 ± 13 years-old, 57% male). Median plasma malondialdehyde (MDA) was 2.55 (IQR, 1.92–3.66) µmol/L. During a median follow-up of 5.3 (IQR, 4.6–6.0) years, 56 (11%) RTR developed NODAT. In Cox proportional-hazards regression analyses, MDA was inversely associated with NODAT, independent of immunosuppressive therapy, transplant-specific covariates, lifestyle, inflammation, and metabolism parameters (HR, 0.55; 95% CI, 0.36–0.83 per 1-SD increase; *p* < 0.01). Dietary antioxidants intake (e.g., vitamin E, α-lipoic acid, and linoleic acid) were effect-modifiers of the association between MDA and NODAT, with particularly strong inverse associations within the subgroup of RTR with relatively higher dietary antioxidants intake. In conclusion, plasma MDA concentration is inversely and independently associated with long-term risk of NODAT in RTR. Our findings support a potential underrecognized role of oxidative stress in post-transplantation glucose homeostasis.

## 1. Introduction

New-onset diabetes after transplantation (NODAT) is a major metabolic complication of solid organ transplantation, with a reported incidence of up to 50% [1]. Consequences of NODAT are detrimental for renal transplant recipients (RTR) as it is associated with reduced recipient survival, increased rate of cardiovascular events, and impaired graft survival in the long term [2,3]. In the era of high-dose steroid regimens, twelve-months cumulative incidence of NODAT was significantly higher, and the main risk factor identified for the occurrence of NODAT was immunosuppressant therapy [3]. However, with new cyclosporine-based and tacrolimus-based regimens [1,4], it is well-documented that the largest number of incident cases of NODAT occurred, indeed, after the first year of transplantation [4,5], and other agents potentially involved in the long-term pathogenesis of the disease remain to be elucidated. In order to improve the outcomes of RTR, it is of great interest to know which factors contribute to this long-term NODAT development and maintenance [2].

Oxidative stress (OS) is a factor that in different studies has been linked with both physiological response to insulin and pathophysiological mechanisms of, e.g., diabetes mellitus; it is also known to be enhanced in RTR when compared to general population [6]. Higher levels of OS biomarkers, e.g., MDA [7], have been found in patients with established diabetes mellitus compared to healthy controls [8], and in patients with diabetes-associated complications compared to patients with noncomplicated diabetes [9]. However, a developing body of evidence has linked oxidative species with insulin signaling [10,11,12,13], and it has been postulated that diabetes mellitus is—to a considerable extent—caused by a failure of the organism to create enough oxidative redox potential [14]. There is, nevertheless, only limited data relating OS with insulin resistance in prediabetes states [15]. Furthermore, to the extent of our knowledge, no longitudinal studies have aimed to study the association between OS biomarkers and long-term incidence of diabetes, which makes it difficult to foresee whether OS biomarkers may prospectively be associated with positive or negative outcomes regarding glucose metabolism outcomes.

In post-transplantation setting, less evidence is available regarding the role of OS on glucose homeostasis. Indeed, the long-term prospective association of systemic OS and the development of NODAT has not been explored. The primary objective of the present study was set to test the hypothesis that post-transplantation OS is associated with the development of NODAT. Furthermore, by considering evidence reporting an effect of dietary antioxidant intake on the development of type 2 diabetes and NODAT [16,17], we aimed to assess whether the potential association of MDA with NODAT may be modified by regular dietary antioxidant fatty acids intake. Finally, we investigated whether OS is associated with the secondary end-points of long-term all-cause mortality, cardiovascular mortality, and graft failure.

## 2. Materials and Methods

### 2.1. Study Design and Patient Population

In this prospective cohort study, all adult RTR with a functioning graft for at least one year who visited the outpatient clinic at the University Medical Center of Groningen (The Netherlands) between November 2008 and May 2011 were considered eligible to participate. Baseline data was obtained at least one year after transplantation with a median of five years. We excluded RTR with diabetes mellitus at baseline or before transplant (defined as fasting plasma glucose ≥126 mg/dL (7.0 mmol/L) and/or use of glucose lowering drugs) (*n* = 173); also patients who underwent combined pancreas-kidney transplantation (*n* = 5) or whose plasma MDA concentration measurement at baseline was missing (*n* = 12), resulting in 516 RTR eligible for statistical analyses. The patients were followed-up until 1 April 2014. Collection of these data was ensured by the continuous surveillance system of the outpatient clinic of our university hospital and close collaboration with affiliated hospitals. Follow-up was performed according to the guidelines of the American Society of Transplantation [18].

The primary end-point of the current study was the long-term development of NODAT. Secondary end points were all cause-mortality, cardiovascular mortality, and graft failure. No participants were lost due to follow-up. The current study was approved by the institutional review board (METc 2008/186) and adhered to the Declarations of Helsinki and Istanbul.

### 2.2. Data Collection

Baseline data was collected during a visit to the outpatient clinic, following a detailed protocol described elsewhere [19]. Anthropometric measurements were taken while participants wore indoor clothing without shoes. Systolic blood pressure (SBP) and diastolic blood pressure (DBP) were measured using a semiautomatic device (Dinamap1846; Critikon, Tampa, FL, USA) every minute for 15 min, following a strict protocol as described before [20].

Three questionnaires were administered to patients: first, the Short QUestionnaire to ASsess Health-enhancing physical activity (SQUASH) score for information about the daily physical activity [21]. Second, a questionnaire regarding smoking behavior to classify patients as current, previous or never smokers. Third, a semiquantitative self-administered food frequency questionnaire (FFQ) of 177 items to collect information on dietary intake during the past month. The FFQ was developed at Wageningen university, previously validated for our population, and it has been updated several times [22]. Number of servings was recorded in natural units (e.g., slice of bread) or household measures (e.g., a teaspoon). Subsequently, all dietary data were converted into total energy and nutrient intake per day, using the Dutch Food Composition Table 2006 [23]. Specific nutrient intakes were adjusted for total energy intake according to the residual method [24].

Of note, except for discouraging excess sodium intake and encouraging weight loss in overweight individuals, no specific dietary counseling was included, nor was dietary recommendation regarding antioxidant fatty acids intake or supplementation advised to the study subjects. Other relevant recipient and transplant information was extracted from the Groningen Renal Transplant Database, as described in detail before [25].

### 2.3. Measurements and Definitions

Fasting blood samples and complete 24-hour urine collection were taken at baseline. Serum creatinine was determined by using the Jaffe reaction (MEGA AU510; Merck Diagnostica, Darmstadt, Germany); plasma glucose by the glucose oxidase method (YSI 2300 Stat Plus; Yellow Springs Instruments, Yellow Springs, OH, USA); total cholesterol by the cholesterol oxidase-phenol aminophenazone method (MEGA AU510); HDL cholesterol by the cholesterol oxidase-phenol aminophenazone method on a Technicon RA-1000 (Bayer Diagnostics, Mijdrecht, the Netherlands); and plasma triglycerides by the glycerol-3-phosphate oxidase-oxidase method (YSI 2300 Stat Plus). LDL cholesterol was calculated by using the Friedewald equation; estimated glomerular filtration rate (eGFR) by the serum creatinine based Chronic Kidney Disease EPIdemiology collaboration equation (CKD-EPI) [26]; and the cumulative dose of prednisolone as the sum of the maintenance dose of prednisolone from transplantation until baseline. Plasma MDA concentration was chosen as the biomarker of OS because it has been used before in studies regarding pathologies of the glucose metabolism [8,9]; it was measured by high-performance liquid chromatography with a photodiode array detector as described by Faizan et al. to improve the sensitivity offered by spectrophotometrically methods [27].

NODAT was defined according to the International Expert Panel recommendations based on the 2003 American Diabetes Association criteria [28] and the HbA1c criterion proposed by the International Expert Panel of the international consensus meeting on post transplantation diabetes mellitus [29]. The diagnosis was made with the fulfillment of one or more of the following: symptoms of diabetes (classic symptoms, including polyuria, polydipsia, and unexplained weight loss) plus random plasma glucose concentration ≥200 mg/dL (11.1 mmol/L); fasting plasma glucose ≥126 mg/dL (7.0 mmol/L); plasma HbA1c ≥ 6.5%; or use of glucose-lowering medication. If fasting plasma glucose was elevated, a confirmatory laboratory test was performed, after which the diagnosis of NODAT was made.

Cardiovascular death was defined as the principal cause of death being cardiovascular in nature (International Classification of Diseases (ICD)-9 codes 410–447). The cause of death was obtained by linking the number of the death certificate to the primary cause of death as coded by a physician from the Central Bureau of Statistics according to the ICD-9 [30]. Graft failure was defined as restart of dialysis or retransplantation.

### 2.4. Statistical Analyses

Data analyses, computations, and graphs were performed with SPSS 22.0 software (IBM Corporation, Chicago, IL, USA), R version 3.2.3 software (The R-Foundation for Statistical Computing, Vienna, Austria), and GraphPad Prism version 7 software (GraphPad Software, San Diego, CA, USA).

For descriptive statistics data are presented as mean ± standard deviation (SD) for normally distributed data, and as median (interquartile range (IQR)) for variables with a non-normal distribution. Categorical data are expressed as number (percentage). Crude and age, sex, and eGFR-adjusted linear regression analyses were performed to examine the association of baseline characteristics with circulating MDA. Residuals were checked for normality and natural log-transformed when appropriate. In order to study in an integrated manner which baseline variables were independently associated with and were determinants of circulating MDA, we performed stepwise backwards multivariable linear regression analyses. For inclusion and exclusion in these analyses, *p*-values were set at 0.2 and 0.05, respectively.

NODAT development was visualized by Kaplan–Meier curves according to tertiles of plasma MDA concentration, with statistical significance among curves tested by log-rank (Mantel–Cox) test. The prospective association of plasma MDA concentration with the different outcomes was assessed through Cox regression analyses. We first performed crude analyses followed by additive adjustments for demographic and anthropometric factors (age, sex, and BMI) in model 1; metabolism-related variables (glucose, HbA1c, and HDL cholesterol) in model 2; lifestyle characteristics (current smoking, alcohol intake, and SQUASH score) in model 3; transplantation-related data (transplant vintage and eGFR) in model 4; immunosuppressive therapy (prednisolone dose and use of calcineurin inhibitors) in model 5; and inflammation (high sensitivity C-reactive protein (hs-CRP)) in model 6. NODAT and graft failure were censored at the date of last follow-up or death. Models were checked for the fulfillment of the assumptions of Cox regression analysis. The assumptions were met.

Furthermore, we performed prespecified analyses in which we tested for potential effect-modification by dietary intake of antioxidant fatty acids using multiplicative interaction terms over the fully adjusted model. In case of significant effect-modification, we proceeded with stratified prospective analyses for the concerned variable. Cut-off points of originally continuous variables used in the stratified analyses were determined so they would allow for an as much as possible similar number of events in each subgroup, and thus allow for similar statistical power for the assessment of the primary association under study (MDA concentration and NODAT) in each subgroup after stratification of the overall population. Since the number of events was reduced in each subgroup these analyses were adjusted analogous to model 3 of the overall prospective analyses to avoid overfitting. Also, since the dietary intake of antioxidant fatty acids could also be a potential confounder, we investigated if adjusting for this variable changed the association between MDA and NODAT.

For all statistical analyses, a statistical significance level of *p* ≤ 0.05 (two-tailed) was used, except for the effect-modification analyses where the significance level was *p* ≤ 0.1 (two-tailed) [31].

## 3. Results

### 3.1. Baseline Characteristics

In total 516 RTR (57% men) were included in the analyses with a mean ± SD age of 51 ± 13 years. Patients were included at a median of 5.2 (IQR 2.0–12.2) years after transplantation. The median plasma MDA concentration was 2.55 (IQR 1.92–3.66) µmol/L. Baseline characteristics of the overall RTR population are shown in Table 1. In crude linear regression analyses, glucose concentration had a significant direct association with plasma MDA concentration (*β* = 0.10, *p* = 0.02), which was not modified after adjustment for age, sex, and eGFR. Other variables with significant associations with plasma MDA concentration after adjustment were eGFR (*β* = 0.10, *p* = 0.03) and leucocytes concentration (*β* = 0.10, *p* = 0.03). A final reduced model of baseline variables obtained through backwards linear regression analyses (*α* = 0.05) included glucose concentration (*β* = 0.11, *p* = 0.02), eGFR (*β* = 0.08, *p* = 0.09) leucocytes concentration (*β* = 0.10, *p* = 0.03), HDL concentration (*β* = 0.10, *p* = 0.04), and alcohol intake (*β* = −0.09, *p* = 0.07) (Table 1).

### 3.2. Prospective Analyses on NODAT

During a median follow-up of 5.3 (IQR 4.6–6.0) years, NODAT developed in 56 (11%) RTR. Kaplan–Meier curves for NODAT development by tertiles of RTRs according to circulating MDA are shown in Figure 1. NODAT distribution was significantly different according to the log-rank test (*p* = 0.02). Cox regression analyses showed that plasma MDA concentration is inversely associated with the risk of NODAT (HR, 0.61; 95% CI, 0.41–0.92 per 1-SD; *p* = 0.02). This association was independent of adjustment for demographic and anthropometric factors, metabolism-related variables, lifestyle factors, transplantation-related data, immunosuppressive medication, and inflammation (HR, 0.55; 95% CI, 0.36–0.83 per 1-SD; *p* < 0.01) (Table 2).

### 3.3. Secondary Analysis on MDA and NODAT

In effect-modification analyses, we found that the association between MDA and the risk of NODAT was significantly modified by vitamin E, linoleic acid (LA), and α-lipoic acid (ALA) intake in regular diet (*p*_interaction_ = 0.06, 0.02, and 0.02; respectively). Thus, we performed stratified prospective analyses by subgroups of RTR according to vitamin E intake (≤ or >13.64 mg/day), LA intake (≤ or >14 g/day) and ALA intake (≤ or > or 1.24 g/day), in which cut-off points were determined so they would allow an as much as possible similar number of events in each subgroup. In each subgroup we assessed the association of MDA with development of NODAT and found that MDA was significantly inversely associated with the risk of NODAT in RTR with vitamin E intake >13.6 mg/day (HR, 0.52; 95% CI, 0.29–0.94 per 1-SD; *p* = 0.03), LA intake >14g/day (HR, 0.49; 95% CI 0.28–0.86 per 1-SD; *p* = 0.01), or ALA intake >1.24g/day (HR 0.42, 95% CI, 0.23–0.76 per 1-SD; *p* < 0.01), but not in the subgroups of relatively low intake (Figure 2).

Further, we performed Cox regression analyses with adjustment for these variables to explore if they might also be potential confounders. The association between MDA and NODAT was not significantly modified by additional adjustment for vitamin E intake (HR, 0.52; 95% CI, 0.34–0.81 per 1-SD; *p* < 0.01), ALA intake (HR, 0.55; 95% CI, 0.36–0.83 per 1-SD; *p* < 0.01), or LA intake (HR, 0.56; 95% CI, 0.37–0.84 per 1-SD; *p* < 0.01).

### 3.4. Prospective Analysis on All-Cause Mortality, Cardiovascular Mortality, and Graft Failure

During the same median follow up of 5.3 (IQR 4.6–5.9) years, 86 (17%) RTRs died, 29 (6%) from cardiovascular cause and 57 (11%) developed graft failure. In crude Cox regression analysis, plasma MDA concentration was not significantly associated with the risk of all-cause mortality (HR, 0.96; 95% CI, 0.73–1.25 per 1-SD; *p* = 0.74), cardiovascular mortality (HR, 0.81; 95% CI, 0.58–1.13 per 1-SD; *p* = 0.21), nor death-censored graft failure (HR, 0.89; 95% CI, 0.65–1.23 per 1-SD; *p* = 0.49). Further adjustments did not materially change these findings (Appendix A).

## 4. Discussion

In a large cohort of stable RTR, we showed first that plasma MDA is directly associated with plasma glucose concentration. Second, plasma MDA is inversely associated with long-term risk of NODAT. This association remained present independent of potential confounders, including BMI, baseline glucose concentration and immunosuppressive therapy. Daily dietary intake of antioxidant fatty acids, e.g., vitamin E, LA and ALA were a significant effect-modifier of this association. No association was found whatsoever between MDA and all-cause mortality, cardiovascular mortality, or graft failure. These findings agree with developing evidence that proposes that oxidative status plays an important role in glucose homeostasis [10,11,12,13].

Experimental work has shown that reactive oxygen species (ROS) are part of intracellular insulin signal transmission [10,11]. ROS are upregulated in response to insulin and help to further up-regulate glucose-metabolism associated pathways related, e.g., with insulin-induced aerobic glycolysis [13]. Furthermore, ROS seem to have an effect on enzymes essential for catalytic activity, increasing glucose intake by skeletal muscle cells and glucose transport in adipocytes [11]. Also, human studies have shown that: (i) patients with severe deficiency of plasmatic antioxidants maintain supranormal insulin sensitivity, compared to healthy subjects, even if they are obese [12] and (ii) antioxidant molecules supplementation abrogates the usually generated increase in insulin sensitivity of patients on exercise interventions [32]. The current study, performed in a high-risk of new-onset diabetes population, provides for the first-time prospective evidence in line with aforementioned basic studies, and may further support the postulate of James Watson, according to which, diabetes mellitus may be caused by an incapacity of the cell to produce an oxidative redox environment [14].

Controversy may arise from data that has shown higher plasma MDA concentration in patients with diabetes mellitus than in healthy controls [8], and in patients who develop diabetes-related complications than in those without them [9]. However, it is known that ROS—as intracellular messengers—can generate opposite cellular effects. ROS can activate specific pathways whose products interfere with insulin signaling, e.g., the activation of the redox-sensitive nuclear factor-kappa beta (NF-ĸB) leads to the expression of cytokines such as tumoral necrosis factor α (TNF-α), and interleukins (ILs) such as IL-1β and IL-6 and all these products have a quenching effect on insulin signaling [11]. On the other hand, as mentioned before, ROS can activate signaling pathways important to fulfil insulin functions. Also, they are known to be themselves and stimuli to increase cellular antioxidant capacity [33] by the activation of specific response elements known as Nuclear factor-erythroid related factor 2- antioxidant response elements (Nrf2-ARE); this induction of endogenous antioxidant mechanisms by ROS has been specifically named mitochondrial hormesis and has gained interest in the last years [34], as it is proposed that, contrary to traditional thinking, ROS are not merely deleterious but they are necessary to reach oxidative balance inside the cell. Intensity, location, duration, and concentration of the oxidant stimulus seem to be crucial in defining whether ROS have a physiological or a pathological outcome. However, the specific thresholds that spawn the differential responses have not been determined yet [10,11]. This also might be a potential explanation of why studies regarding antioxidant supplementation have not shown to be beneficial in RTR [35], and why we did not find an association between OS and mortality, cardiovascular mortality, or graft failure.

Our data also provided evidence that LA, ALA, and vitamin E intake modify the association between MDA and NODAT. Conceivable interpretations of these findings are as follows: MDA is formed after the peroxidation of double bonds of unsaturated fatty acids such as LA [7]. Food containing important amounts of unsaturated fatty acids usually also contain substantial amounts of vitamin E, which prevents them from rancidification [36]. It is possible that in this context, high MDA concentration is a marker of a diet rich in antioxidants and unsaturated fatty acids, which has been suggested to reduce diabetes incidence [16,37]. However, when we adjusted for intake of these nutrients to evaluate them as potential confounders, the association between MDA and NODAT remained materially unaltered. An alternative explanation might be the aforementioned Nrf2-ARE pathway. This pathway has been of particular interest in the study of antioxidant molecules as therapeutic interventions, because previous authors have proposed that these interventions could show beneficial results if they were combined with unsaturated fatty acids as precursors of oxidative stress [38]. The rationale is that through Nrf2-ARE pathway activation, provision of pro-oxidant and antioxidants agents would trigger the antioxidant cellular defenses [33], thus yielding cell precondition to new oxidative challenges [39], and ultimately allowing cells to reach hormesis. This fits with our findings that high levels of MDA, although significantly inversely associated with NODAT in all our population, showed a stronger association in the patients with relatively higher intake of antioxidant in regular diet according to our subgroup analyses. Our findings might also support previous suggestions of potential protector effect of antioxidant-rich diets against NODAT [17].

The present study has several strengths. To the extent of our knowledge, it comprises the largest cohort of patients at risk of new-onset diabetes after transplantation in which the relationship between oxidative stress and NODAT has been evaluated. Moreover, our extensively phenotyped cohort allowed us to control for several potential confounders, among which anthropometric measurements, smoking status, baseline glucose metabolism markers, and immunosuppressive therapy were accounted for. Furthermore, NODAT cases were diagnosed according to International Expert Panel recommendations that were based on American Diabetes Association criteria [28], which agrees with usual clinical practice in transplant centers. Another strength of the study is that we included only stable RTRs who were 1-year post-transplantation, resulting in exclusion of transient post-transplantation hyperglycemia in NODAT diagnosis. Hyperglycemia is extremely common in the early posttransplant period and can occur as a result of rejection therapy, infections, and other critical conditions. Therefore, the formal diagnosis of NODAT in RTRs should only be based on likely maintenance of immunosuppression, stable kidney function, and absent acute infections [29]. The present study also has several limitations. It was carried out in a center with over-representation of Caucasian population, which calls prudence to extrapolation of our results to populations of other ethnicities. Another limitation of our study is that we only measured MDA concentrations in baseline samples. Most epidemiological studies use a single baseline measurement to predict outcomes, which adversely affects predictive properties of variables associated with outcomes. If intraindividual variability of predictive biomarkers is taken into account, this results in strengthening of predictive properties that, despite sometimes containing considerable intraindividual day-to-day variation, also existed for single measurements of these biomarkers [40,41]. The higher the intraindividual day-to-day variation is, the greater one would expect the benefit of repeated measurement for prediction of outcomes [40,41]. Next, although MDA has been the most commonly used OS biomarker in studies regarding glucose metabolism [9,10], further studies may want to account for other OS biomarkers to further validate our findings. Finally, the observational nature of this study makes it difficult to discern whether high levels of MDA are protective against NODAT or merely a marker of lower risk for NODAT; and, as with any observational study, residual confounding may have existed despite the substantial number of potentially confounding factors for which we adjusted, including well identified risk factors for NODAT.

In conclusion, plasma MDA concentration is inversely and independently associated with long-term risk of NODAT in stable RTR. This study provided for the first time relevant prospective data on the role of oxidative stress on glucose metabolism in a high-risk of diabetes population. This may further support already published basic studies and further promote studies to widen our knowledge on the role of oxidative stress in the pathophysiological mechanisms leading to diabetes and NODAT, which might be of relevant use in exploring novel therapeutic approaches to prevent and treat NODAT; also, it indicates that studies exploring antioxidant supplementation in RTR should explore and report metabolic outcomes in the long-term.

## Figures and Tables

**Figure 1 jcm-08-00453-f001:**
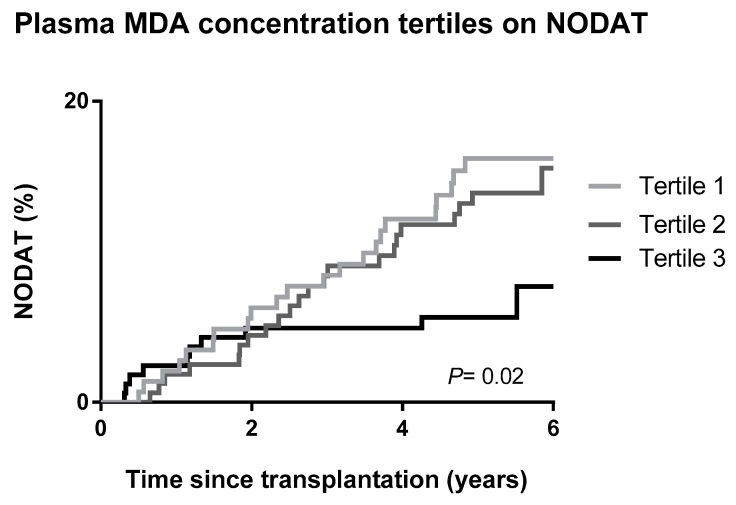
Kaplan–Meier curves for NODAT according to tertiles of plasma MDA concentration in RTR. Tertile 1: <2.15 µmol/L; Tertile 2: 2.15–3.09 µmol/L; Tertile 3: >3.09 µmol/L. *p* value was calculated by Log-rank (Mantel cox) test.

**Figure 2 jcm-08-00453-f002:**
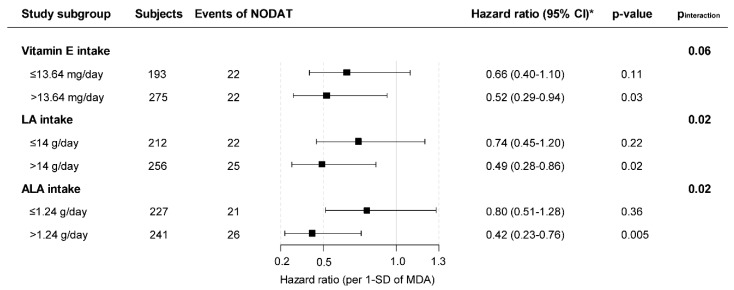
Stratified analysis of the association of plasma MDA concentrations with NODAT. * For the association between MDA and NODAT. HR are reported per 1-SD increase in plasma MDA concentration. Nutrient intake was adjusted for total energy intake according to the residual method. HR adjusted for age, sex, BMI, plasma glucose, HbA1c, smoking status, alcohol intake, and SQUASH score are shown.

**Table 1 jcm-08-00453-t001:** Baseline characteristics of the study population and its association with circulating malondialdehyde (MDA) (*n* = 516).

Baseline Characteristics			Plasma MDA, Ln		
			Linear Regression ^¥^	Adjusted Linear Regression ^†^	Backwards Linear Regression ^§^
			Std. β	Std. β	Std. β
Plasma MDA, µmol/L		2.55 (1.92–3.66)	–	–	–
Demographic and anthropometric					
	Age, years	52 ± 13	0.01	0.02	
	Male sex, *n* (%)	292 (57)	−0.06 *	−0.07 *	~
	Weight, kg	79.0 ± 15.4	−0.04	−0.01	
	Height, cm	174 ± 10	−0.04	0.02	
	BMI, kg/m^2^	26.0 ± 4.4	−0.03	−0.02	
	Waist, cm ^a^	96.4 ± 13.7	−0.07 *	−0.05	
Glucose and lipids metabolism					
	Glucose, mmol/L(mg/dL) ^b^	5.16 (93) ± 0.64 (11)	0.10 **	0.11 **	0.12 **
	HbA1c, % ^c^	5.67 ± 0.36	0.05	0.05	
	Impaired fasting glucose, *n* (%)	122 (24)	0.06 *	0.07 *	~
	Total cholesterol, mmol/L	5.12 ± 1.11	0.05	0.05	
	HDL cholesterol, mmol/L ^d^	1.3 (1.1–1.7)	0.09 **	0.06 *	0.09 **
	LDL cholesterol, mmol/L ^d^	3.0 ± 0.9	−0.04	−0.04	
	Triglycerides, mmol/L ^e^	1.62 (1.21–2.16)	0.03	0.05	
Transplantation-related data					
	Time after transplant, years	5.2 (2.0–12.2)	0.03	0.02	
	Living donor, *n* (%)	187 (36)	0.07 *	0.07 *	~
	Pre-emptive, *n* (%)	92 (18)	0.03	0.02	
Immunosuppressive therapy					
	Acute rejection treatment, *n* (%)	124 (24)	0.06 *	0.08 *	~
	Use of calcineurin inhibitors				
	Tacrolimus, *n* (%)	89 (17)	−0.01	0.02	
	Cyclosporine, *n* (%)	194 (38)	−0.02	−0.01	
	Use of proliferation inhibitors				
	Azathriopine, *n* (%)	95 (18)	0.01	0.01	
	Mycophenolic acid, *n* (%)	340 (66)	0.03	0.03	
	Prednisolone cumulative dose, g	16.9 (5.8–36.3)	0.02	0.02	
Cardiovascular history					
	History of CV disease, *n* (%) ^f^	204 (40)	0.01	0.01	
	SBP, mmHg ^e^	135 ± 17	−0.03	−0.01	
	DBP, mmHg ^e^	83 ± 11	0.05	0.08 *	~
	Use of antihypertensive medication, *n* (%)	448 (87)	−0.05	−0.02	
Graft function and inflammation					
	Serum creatinine, µmol/L ^d^	123 (100–159)	−0.07 *	0.12 *	~
	eGFR (CKD-EPI), mL/mind ^d^	53 ± 20	0.10 **	0.10 **	~
	Protein excretion, g/day	0.18 (0.02–0.32)	0.01	0.04	
	hs-CRP, mg/L ^g^	1.4 (0.6–3.8)	<0.01	<0.01	
	Leucocytes, × 10^9^/L ^e^	7.8 (6.3–9.6)	0.10 **	0.09 **	0.12 **
Nutrition					
	Plasma albumin, g/L ^d^	43.3 ± 3.0	0.002	−0.003	
	Kcal intake, kcal/day ^h^	2189 ± 617	−0.002	0.01	
	Fatty acids intake ^h^				
	n-6 LA, g/day ^∧^	15 (13–19)	0.03	0.05	
	n-6 AA, g/day ^∧^	0.05 (0.04–0.06)	0.02	0.02	
	n-3 ALA, g/day ^∧^	1.25 (1.02–1.60)	0.01	0.03	
	n-3 EPA, g/day ^∧^	0.04 (0.01–0.09)	0.05	0.05	
	n-3 DHA, g/day ^∧^	0.06 (0.03–0.13)	0.06	0.06	
Lifestyle					
	Current smokers, *n* (%) ^i^	67 (13)	−0.01	0.002	
	Alcohol intake, g/day ^h^	2.92 (0.04–11.52)	−0.08 *	−0.08 *	~
	SQUASH-score, intensity × hours	5555 (2640–8513)	−0.02	<0.01	

* *p* value < 0.20; ** *p* value < 0.05. ^¥^ Crude linear regression analysis. ^†^ Linear regression analysis adjusted for age, sex, and eGFR. ^§^ Stepwise backwards linear regression analysis; for inclusion and exclusion in this analysis, *p* Values were set at 0.2 and 0.05, respectively. ~ Excluded from the final model. Data available in: ^a^ 499, ^b^ 514, ^c^ 495, ^d^ 455, ^e^ 515, ^f^ 398, ^g^ 484, ^h^ 468, ^i^ 490 patients. MDA, malondialdehyde; Std. β, standarized B coefficient; eGFR, estimated glomerular filtration rate; CV, cardiovascular; HbA1c, glycosylated hemoglobin; hs-CRP, high-sensitive C-reactive protein; kcal, kilocalories; LA, linoleic acid; AA, arachidonic acid; ALA, α-lipoic acid; EPA, eicosapentaenoic acid; DHA, docosahexaenoic acid. ^∧^ Adjusted for total caloric intake according to the residual method.

**Table 2 jcm-08-00453-t002:** Plasma MDA concentration and new-onset diabetes after transplantation (NODAT) in renal transplant recipients (RTR, *n* = 516).

NODAT	HR (95% CI) Per 1-SD	*p*
Crude model	0.61 (0.41–0.92)	0.02
Model 1	0.63 (0.42–0.94)	0.02
Model 2	0.54 (0.36–0.83)	<0.01
Model 3	0.54 (0.35–0.82)	<0.01
Model 4	0.56 (0.37–0.85)	<0.01
Model 5	0.55 (0.36–0.83)	<0.01
Model 6	0.55 (0.36–0.83)	<0.01

In total, 56 (11%) RTR developed NODAT. Model 1: crude model plus adjustment for demographic and anthropometric characteristics. Model 2: model 1 plus adjustment for metabolism-related variables. Model 3: model 2 plus adjustment for lifestyle characteristics. Model 4: model 3 plus adjustment for transplantation-related data. Model 5: model 4 plus adjustment for immunosuppressive therapy. Model 6: model 5 plus adjustment for inflammation.

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
