# Peer review of "Plasma Malondialdehyde and Risk of New-Onset Diabetes after Transplantation in Renal Transplant Recipients: A Prospective Cohort Study"

_jcm, 2019, doi:10.3390/jcm8040453_

Round 1
Reviewer 1 Report
Comments to jcm-456907
The authors reported a potential under-recognized role of oxidative stress in post-transplantation glucose homeostasis. They found that plasma MDA concentration is inversely and independently associated with long-term risk of NODAT in RTR. There are some concerns should be clarified.
1.The authors only showed the initial blood MDA level. The MDA level could be flocculate during the follow-up periods, average 5.3 years. It is very important to know if those patients with NODAT are continuously high MDA level.
2.If the intervention by vitamin E, LA and ACA will change the level of MAD. The authors should analyze the change of MDA after intervention in their cohort.
3.In P2, L54, the author should use the full name of “OS”, when this term is the first time to appear in this manuscript.
Author Response
Manuscript ID: jcm-456907
Detailed, itemized response to the comments of Reviewer #1.
Reviewer #1:
The authors reported a potential under-recognized role of oxidative stress in post-transplantation glucose homeostasis. They found that plasma MDA concentration is inversely and independently associated with long-term risk of NODAT in RTR. There are some concerns should be clarified.
Response: We thank the reviewer for the comments, and we hope that the following discussion clarifies the respective concerns.
Specific comments:
Comment #1. The authors only showed the initial blood MDA level. The MDA level could be flocculate during the follow-up periods, average 5.3 years. It is very important to know if those patients with NODAT are continuously high MDA level.
Response: We appreciate the comment of the Reviewer. We would like to kindly make notice that most epidemiological studies use a single baseline measurement to predict outcomes, which adversely affects predictive properties of variables associated with outcomes. Likewise, regrettably, only baseline MDA measurements are available in the current study. Nonetheless, we would like to stress on that the higher the intra-individual day-to-day variation of a biomarker would be, the greater one would expect the benefit of repeated measurement for prediction of outcomes. Hence, the use of a single measurement of MDA actually leads to underestimation of the underlying true effect in such a way that repeated MDA measurements would potentially strengthen the prospective association hereby reported with NODAT. To accommodate this comment of the Reviewer, we added the corresponding discussion to the limitations paragraph (L379-386), in which we provide reference to the following bibliography (references 41-42):
- Koenig W., et al. Refinement of the Association of Serum C-Reactive Protein Concentration and Coronary Heart Disease Risk by Correction for within-Subject Variation Over Time: The MONICA Augsburg Studies, 1984 and 1987. Am. J. Epidemiol. 2003; 158: 357–364.
- Danesh J., et al. C-Reactive Protein and Other Circulating Markers of Inflammation in the Prediction of Coronary Heart Disease. N. Engl. J. Med. 2004; 350: 1387–1397.
Comment #2. If the intervention by vitamin E, LA and ACA will change the level of MAD. The authors should analyze the change of MDA after intervention in their cohort.
Response: We thank the Reviewer for this comment. We would like to emphasize, however, that the current is an observational study by design, and that no intervention was performed in our study population. As stated in L114-117 of methods section, no dietary recommendations nor nutrient supplementation were given to any of the study subjects. Instead we analysed the effect-modification by regular dietary intake of the aforementioned antioxidant nutrients on the primary prospective association under study (i.e., MDA with long-term occurrence of NODAT). To accommodate the comment of the Reviewer, we make this point more explicit by replacing the term “dietary intake” by “regular dietary intake” in the revised version of the manuscript.
Comment #2. In P2, L54, the author should use the full name of “OS”, when this term is the first time to appear in this manuscript.
Response: We thank the Reviewer for noting this. Accordingly, we have replaced the abbreviation OS in L56.
Reviewer 2 Report
This is a provocative study. The study is however very heavy on statistics.
The conventional wisdom is that high levels of Oxidative Stress biomarkers such as MDA is high in diabetes.
This study offers a different perspective on MDA level and the risk of NODAT.
The dietary aspects I'm assuming have been validated previously in this population. That is not clear.
I was confused with figure 2. With the ALA intake, the percentage of events with the low intake and hlgh intake we're 9.2% and 10.7%; yet the hazard ratio is smaller for the high intake group.
Also need validation as to how those specific cut off values we're chosen for the dietary supplements.
The controversial finding of lower MDA with higher risk of NODAT needs to be explained better and in a more simplistic way.
This also begs the fact if MDA is the ideal marker to be studied.
Nonetheless, a controversial finding that needs to be explored further.
Author Response
Manuscript ID: jcm-456907
Detailed, itemized response to the comments of Reviewer #2
Reviewer #2:
This is a provocative study. The study is however very heavy on statistics. The conventional wisdom is that high levels of Oxidative Stress biomarkers such as MDA is high in diabetes. This study offers a different perspective on MDA level and the risk of NODAT.
Response: We thank the Reviewer for the kind appraisal of our manuscript, and we hope that the clarifications provided below further support our findings. We agree with the reviewer regarding conventional wisdom about oxidative stress in diabetes. Accordingly, we referred to studies that report higher levels of oxidative stress biomarkers in populations with diabetes when compared to general population (L58-61). In pre-diabetic states, however, this association is less clear and, indeed, previous studies have reported a potential regulatory role of oxidative stress in glucose metabolism, which is consistent with the physiological regulatory direction of our findings. References 11 to 14 specify basic studies in which a role of oxidative stress in insulin signalling pathways was found. In reference 15, we refer to the postulate by Professor James Watson, in Lancet, on Diabetes Mellitus as a redox disease caused by an incapacity of the organism to produce enough oxidative potential. Furthermore, it has been theorized that the decrease of antioxidant defences in the B-cells of the pancreas constitutes an evolutionary adaptation (Rashidi, A; et al. Metabolic evolution suggests an explanation for the weakness of antioxidant defences in beta-cells. Mech Ageing Dev 2009; 130(4): 216-21). Hence, a growing body of evidence point towards a potential regulatory role of oxidative stress in glucose metabolism. Our study provides clinical data in line with that evidence.
Specific comments:
Comment #1. The dietary aspects I'm assuming have been validated previously in this population. That is not clear.
Response: We thank the Reviewer for this comment. Indeed, all information regarding diet was obtained through a food-frequency questionnaire validated for our population. The validation process was published in the American Journal of Clinical Nutrition. To improve the transparency of this point, in the materials and methods section, where we describe the use of the food-frequency questionnaire, we have now added that this tool has previously been validated for our population and we provide bibliographical reference to the original article of the validation process (L108-L109).
Comment #2. I was confused with figure 2. With the ALA intake, the percentage of events with the low intake and high intake we're 9.2% and 10.7%; yet the hazard ratio is smaller for the high intake group.
Response: We appreciate this comment of the Reviewer. We would like to kindly underline that in figure 2 we are not exploring the association between ALA, LA or vitamin E with NODAT, instead we are performing stratified analyses of the primary association (i.e., MDA and long-term occurrence of NODAT). These stratified analyses were performed within independent subgroups of RTR according to regular dietary intake of the 3 aforementioned antioxidant nutrients. Thus, this figure do not represent the analyses of relative risk of one group against the other. Instead, within each subgroup of study subjects (e.g., RTR with relatively higher ALA intake) we hereby analysed the prospective association between MDA and NODAT, by means of Cox proportional-hazards regression, with the continuous variable MDA concentration as the predictor. Stratified analyses, therefore, are not performed upon reference to the other (remaining) subgroup of patients. In practice, this means that the size of each subgroup, as well as its relative number of events, rather provide a note of the statistical power after stratification of the overall population, but not an inherent link to the prospective association under study in subgroups. To accommodate this comment of the Reviewer, we further clarify that hereby we report the association of MDA concentration with NODAT (instead of the association between ALA, LA and vitamin E with NODAT), by adding in the figure legend the sentence: “HR are reported per unit increase of plasma MDA concentration (µmol/L)” (L262).
Comment #3. Also need validation as to how those specific cut off values we're chosen for the dietary supplements.
Response: We appreciate this comment of the Reviewer. We would like to kindly underline that, as stated in L114-117 of methods section, no dietary recommendations nor nutrient supplementation were given to any of the study subjects, but instead we analysed the effect-modification of regular dietary intake of pre-specified antioxidants nutrients (i.e., vitamin E, LA and ALA) on the primary association under study (i.e., MDA and long-term occurrence of NODAT). Next, cut-off points of regular dietary intake of the pre-specified nutrients were selected in such a way that would allow each subgroup to display a similar number of events, and thus maximize statistical power to explore the primary association in each subgroup. To accommodate this comment of the Reviewer, we extended this explanation in the Materials and Methods section (L183-186).
Comment #4. The controversial finding of lower MDA with higher risk of NODAT needs to be explained better and in a more simplistic way.
Response: We thank the Reviewer for this comment. Accordingly, we rephrased the discussion so it would allow a more simplistic reading of the proposed underlying mechanisms for our findings. We gave particular attention to the paragraph from L300 to L319, in which most of the mechanisms are described, and we believe that with this rewriting, the central idea, which is to show the paradoxical effect of oxidative stress in glucose metabolism, becomes more explicit.
Comment #5. This also begs the fact if MDA is the ideal marker to be studied.
Response: We thank the Reviewer for this comment. It is of our consideration that the most meaningful advantage of plasma MDA as marker of oxidative stress, in this particular context, is that it has previously been used for studies in relation to glucose metabolism (L130-131), which gives our study more comparability with the literature. Plasma MDA has been widely used as a marker of oxidative stress in several studies, and it is currently one of the most commonly used. To support this point, we refer to the follow bibliography:
-Tsikas, D. Assessment of lipid peroxidation by measuring malondialdehyde (MDA) and relatives in biological samples, Analytical and biological challenges. Anal Biochem 2017; 524: 13–30.
-Nielsen, F; et al. Plasma malondialdehyde as biomarker for oxidative stress: reference interval and effects of life-style factors. Clin Chem 1997; 43(7): 1209-14.
Nonetheless, we think that further studies may want to account for other oxidative stress biomarkers to further validate our findings. Accordingly we added commenting on this regard to the limitations paragraph of the discussion in this revised version of the manuscript (L386-389)
Comment #6. Nonetheless, a controversial finding that needs to be explored further.
Response: We thank the Reviewer for the kind appraisal of our conclusion, as we believe precisely that this study, with its seemly controversial finding, opens the door for future clinical research approaches regarding a potential role of oxidative stress in glucose homeostasis.
Round 2
Reviewer 2 Report
I am satisfied with the changes. However, I still need clarification on Figure 2. Particularly this sentence "MDA was significantly inversely associated with the risk of NODAT in RTR with vitamin E intake >13.6 mg/day (HR, 0.52; 95% CI, 0.29–0.94 per SD; P=0.03), LA intake >14g/day (HR, 0.49; 95% CI 0.28– 0.86 per SD; P=0.01) or ALA intake >1.24g/day (HR 0.42, 95% CI, 0.23–0.76 per SD; P<0.01), but="" not="" in="" the="" subgroups="" of="" relatively="" low="" intake="" figure="" i="" am="" assuming="" that="" column="" mean="" number="" cases="" nodat.="" especially="" with="" ala="" high="" events="" appear="" higher="" vs.="" .="" reading="" it="" if="" can="" you="" help="" clarify="" span="" style="margin-bottom:0; text-align:justify; background:#F6F6F6" p="" author="" manuscript="" id:="" itemized="" response="" to="" comments="" reviewer="" satisfied="" changes.="" still="" need="" clarification="" on="" 2.="" particularly="" this="" sentence="" mda="" was="" significantly="" inversely="" associated="" risk="" nodat="" rtr="" vitamin="" e="">13.6 mg/day (HR, 0.52; 95% CI, 0.29–0.94 per SD; P=0.03), LA intake >14g/day (HR, 0.49; 95% CI 0.28– 0.86 per SD; P=0.01) or ALA intake >1.24g/day (HR 0.42, 95% CI, 0.23–0.76 per SD; P<0.01), but not in the subgroups of relatively low intake (Figure 2)"
I am assuming that the column "Events" mean the number cases of NODAT. Especially with ALA high intake cohort, the Events appear higher with higher intake (9.2% vs. 10.7%). Am I reading it wrong? If so, can you help clarify that?
Response: We thank the Reviewer for the kind appraisal of the changes made to our manuscript. As for figure 2, we will gladly clarify the reviewers' concern. The reviewer is right in assuming that “Events” mean the number of cases of NODAT. Because we found significant effect-modification by continuous variables of vitamin E, LA and ALA intake on the primary association of MDA with NODAT, we subsequently proceeded with stratified analyses to investigate associations of MDA with NODAT within subgroups of RTR according to intake of (micro)nutrients. Thus, we studied the association of MDA with NODAT within RTR with either relatively lower or relatively higher intake of vitamin E, LA and ALA. It should be noticed that it is always the association of MDA with NODAT which is investigated within these subgroups. To perform these subgroup analyses, we aimed to split the overall population into subgroups of RTR with low/high intake of the (micro)nutrient of interest, in such a way that would allow us to have a roughly equal number of events in each subgroup in order to provide a roughly equal power to study the association of MDA with NODAT by means of Cox regression analyses. The calculated hazards ratios (95% CI) reported in the figure are, therefore, effect estimates for the associations of MDA with NODAT and not effect estimates of the association of (micro)nutrients intake under study (vitamin E, LA or ALA intake) with NODAT. Hence, the rate of events in each subgroup does not necessarily relate to the risk of events as shown by effect estimates, because the risk in each subgroup depends upon the data on MDA among patients within each subgroup, and how that data associates with events (NODAT), separately, in each subgroup. In stratified analyses, one aims to evaluate how a third variable (in this case; intake of vitamin E, LA and ALA) interacts with the association of a primary association under study (in this case; MDA and NODAT). Thus, in the current study we show that MDA associates more strongly with NODAT within RTR with higher, but not lower intake of the antioxidant (micro)nutrients vitamin E, LA and ALA.
As a summary, the reviewer is correct in assuming that the rate of events is slightly higher in the subgroup of RTR with higher intake of ALA, though it should be realized that, hereby, we are showing the association of MDA with NODAT within subgroups of RTR with relatively higher or lower ALA intake. The variable of interest, i.e., MDA is not significantly associated with risk of NODAT within the subgroup of RTR with relatively lower ALA intake, despite of the fact that there is a slightly higher number of events. This differential behaviour of the association of MDA with NODAT upon changes in the third variable, i.e., ALA intake, is what we are displaying in figure 2 and in the paragraph which the reviewer is referring (L220-230). To accommodate the comment of the reviewer, we modified the description of these effect-modification analyses in the “Methods” section (L167-L178) and in the “Results” section (L220-230) of the revised version of the manuscript. To further accommodate the comment of the reviewer, we also changed “Events” to “Events of NODAT” in the column header of figure 2, and we changed the expression of the legend of the x-axis from “Hazard ratio (per SD)” to “Hazard ratio (per SD of MDA)”, and added an asterisk to the column header “Hazard ratio (95% CI)”, linking it to the asterisk in the figure legend to indicate that it refers to the expression of Hazard ratio per SD of MDA.